# Pregnant and breastfeeding women's prospective acceptability of two biomedical HIV prevention approaches in Sub Saharan Africa: A multisite qualitative analysis using the Theoretical Framework of Acceptability

**Mandeep Sekhon**[1]*, **Ariane van der Straten**[2,3], on behalf of the MTN-041/MAMMA Study Team[¶]

**1** Department of Population Health Sciences, School of Population Health and Environmental Sciences, Faculty of Life Sciences & Medicine, King's College London, London, United Kingdom, **2** Center for AIDS prevention studies, University of California San Francisco, CA, United States of America, **3** Women's Global Health imperative, RTI International, Berkeley, CA, United States of America

¶ Membership of the MTN-041/MAMMA Study team is provided in the Acknowledgments.
* Mandeep.sekhon@kcl.ac.uk

## Abstract

HIV infection during pregnancy and breastfeeding has implications for maternal health. Between May- November 2018, we explored prospective acceptability of two novel HIV Pre-exposure Prophylaxis (PrEP) products, oral pills and vaginal rings, through focus group discussions with 65 pregnant and breastfeeding women in Malawi, South Africa, Uganda, Zimbabwe. Qualitative analysis was completed, guided by the Theoretical Framework of Acceptability (TFA). First, a deductive thematic analysis was applied to relevant coded data, into the seven TFA constructs (Affective Attitude; Burden; Ethicality, Intervention Coherence; Opportunity Costs; Perceived Effectiveness; Self-efficacy). Next, an iterative analysis was completed to generate themes within each of the TFA constructs. Women's positive attitudes towards daily oral PrEP highlighted the familiarity of taking pills, understanding the purpose of taking pills, and the perception that it is an effective method to protect mothers and babies from HIV during pregnancy and breastfeeding. Women emphasized the ease of using the ring given its monthly duration that lowers burden on the user, its discreetness and invisibility once in place. The TFA analysis highlighted how acceptability of both methods could be enhanced by focusing on perceptions of the end users (i.e. the women) and not just the products themselves. This approach provided insights into how to refine the intervention materials and plans for implementation.

## Introduction

Across sub-Saharan Africa prevalence of HIV amongst women remains disproportionately high, with women and adolescent girls accounting for 59% of all new infections [1]. Amongst

**Data Availability Statement:** The minimal data set underlying the results reported in this paper are included as Supporting Information files.

**Funding:** The MTN-041 study was designed and implemented by the Microbicide Trials Network (MTN) funded by the National Institute of Allergy and Infectious Diseases through individual grants (UM1AI068633, UM1AI068615 and UM1AI106707), with co-funding from the Eunice Kennedy Shriver National Institute of Child Health and Human Development and the National Institute of Mental Health, all components of the U.S. National Institutes of Health (NIH). The content is solely the responsibility of the authors and does not necessarily represent the official views of the NIH. The funders had no role in study design, data collection and analysis, decision to publish, or preparation of the manuscript. The specific roles of these authors are articulated in the 'author contributions' section.

pregnant and breastfeeding (P/BF) women in some settings, the incidence rate accounts for up to 30% of new HIV infections [2]. Additionally, HIV infection during pregnancy and breast-feeding has implications for maternal health, increasing the risk of mother to child transmission [3].

To address women's vulnerability to HIV during pregnancy and breastfeeding, evaluating HIV prevention methods that can be initiated and used by the women themselves are important. This includes daily oral Pre- Exposure Prophylaxis (PrEP) [4, 5] and monthly vaginal rings [6, 7]. The Word Health Organisation recommends offering daily Truvada™ or emtricitabine(FTC) / tenofovir disoproxil fumarate (TDF) pills to women at considerable risk of HIV infection [5]. Oral PrEP is safe and effective in P/BF women [8–10]. However, access and availability of oral PrEP for uninfected mothers remains limited in many settings [11, 12].

A vaginal ring releasing the antiretroviral dapivirine offers the advantage to deliver drugs continuously for coital independent use, over a month or more. In Phase 3 trials and open-label extensions, the dapivirine vaginal ring was shown to be safe and effective at preventing HIV in non-pregnant women [13–16] with regulatory approval process underway [17].

Pregnant and lactating women are understudied in prevention trials because of concerns about potential harm to the foetus and baby. Thus, there is little safety or acceptability data for biomedical prevention during pregnancy and lactation periods [6, 18, 19]. This is also the case in low- and middle-income countries, where PrEP interventions are most likely to be implemented [20].

Oral and vaginal PrEP are important breakthroughs for HIV prevention and thus require better understanding of factors influencing attitudes and behaviours to determine their real-world effectiveness [11, 21].

To further explore individual, interpersonal, social and cultural factors that may influence uptake of the vaginal ring and oral PreP, the Microbicide Trial Network (MTN)- funded a multi-site qualitative study, **M**icrobicide/PrEP **A**cceptability among **M**others and **M**ale Partners in **A**frica (MTN-041/MAMMA). Primary findings indicated that there was consensus amongst men and women, that P/BF women are at higher risk of HIV due to their partners infidelity. Both women and men welcomed new methods of prevention [7]. Reported influences on future product use included safety considerations for the mother and child dyad; women having support from their partners, men being involved in the decision making process; women expressing the importance of having support from the wider family on decision to use PrEP methods, and endorsement of products by healthcare professionals [7].

Here, we specifically explore the P/BF women's prospective acceptability of the two novel HIV PrEP products, the daily oral pill and the monthly vaginal ring. Exploring acceptability from the perspective of P/BF women is crucial during the development, implementation and scale up of HIV PrEP in these populations [22, 23]. Formative research exploring prospective acceptability of PrEP can identify product-specific perceptions and preferences, potential barriers to uptake and factors that may encourage or hinder adherence to product use [22, 24]. Ultimately, acceptability research can inform implementation and scale -up by informing PrEP counselling and adherence support strategies for women who need or desire HIV prevention [25].

There is no consensus in the HIV literature or a standard definition of acceptability [26–29]. Prior microbicide acceptability research emphasized the physical attributes (e.g. colour, size, smell) and elicited mainly women's hypothetical intention to use a product [26]. This conceptualisation of acceptability has evolved over time, in which acceptability has been considered a multifactorial construct, which incorporates *"factors and interactions of the product with the user, the sex partner, the environment, and social and cultural norms"* (p.2 [28]). The use of behavioural and social sciences theory, and theoretically relevant constructs has also been

advocated to better understand user-related factors that may influence product acceptability [28].

To advance acceptability research, a recent Theoretical Framework of Acceptability (TFA) has been developed to facilitate both quantitative and qualitative assessments of healthcare intervention acceptability [30, 31]. The TFA was developed by inductively synthesising the findings from an overview of reviews and applying deductive methods to theorise the concept of acceptability [30]. In this research, acceptability was defined as "*a multi-faceted construct that reflects the extent to which people delivering or receiving a healthcare intervention consider it to be appropriate, based on anticipated or experienced cognitive and emotional responses to the intervention*" (p. 1 [30]). The TFA consists of seven component constructs: Affective attitude, Burden, Perceived effectiveness, Ethicality, Intervention coherence, Opportunity costs and Self-efficacy [31]. The TFA has been developed to facilitate both qualitative and quantitative assessments of intervention acceptability across three different timepoints: *prospective acceptability* (before intervention engagement); *concurrent acceptability* (during intervention engagement) and *retrospective acceptability* (after intervention engagement) and provides the key advantage of focusing on the users rather than on the biomedical product itself like previous conceptualizations of product acceptability [26, 28, 32]. Sekhon and colleagues propose the TFA provides an evidence base to inform strategies for enhancing acceptability of health interventions [30].

In this paper, we apply the TFA to complete a secondary analysis of the eight FGDs completed with P/BF as part of the MAMMA study [7]. We explored P/BF women's prospective acceptability of two novel HIV PrEP products, the daily oral pills and the monthly vaginal ring. Specifically, we wanted to understand among P/BF women:

1. Using the TFA constructs to guide our understanding, how acceptable are these PrEP products in the given sample?

2. Based on the TFA, how can the acceptability of either of these PrEP products be enhanced?

## Methods

### Study setting and design

This study was embedded within the Microbicide Trial Network (MTN)- funded Microbicide/ PrEP Acceptability among Mothers and Male Partners in Africa (MTN-041/MAMMA), a multi-site qualitative study to identify individual, interpersonal, social and cultural factors that may affect the uptake by P/BF women in Africa of the monthly dapivirine vaginal ring (ring) an investigational product that has received prequalification from WHO, and is under regulatory consideration in multiple settings [17], and a pill for daily oral pre-exposure prophylaxis (PrEP) approved for use by women globally and endorsed by WHO [5] (Table 1).

Multiple stakeholders were interviewed, including women, across four sites in Blantyre, Malawi; Johannesburg, South Africa; Kampala, Uganda; and Chitungwiza, Zimbabwe, between May–November 2018 [7]. This paper focuses on the eight single sex focus group discussions (FGDs, two per site) completed with P/ BF women (or those recently so), who were recruited independently to join a FGD.

### Ethical approval

The MAMMA study protocol was approved by the Western Institutional Review Board (IRB) and by local IRBs at each of the study sites and was overseen by the regulatory infrastructure of the U.S. National Institutes of Health and the MTN.

**Table 1. Attributes of products for HIV Pre-Exposure Prophylaxis (PrEP).**

| | Daily Oral Pill | Monthly microbicide Vaginal Ring |
|---|---|---|
| Drug | tenofovir disoproxil fumarate/emtricitabine or TDF/FTC (Truvada™) | Dapivirine |
| Route of administration | Oral | Vaginal |
| Frequency of dosing | Daily | Monthly |
| Safety | No safety-related rationale for disallowing or discontinuing PrEP use during pregnancy and breastfeeding.[1] | Safety confirmed in phase 3 HIV prevention trials, and open-label phase 3b trials in non-pregnant women. Limited safety data for pregnant and breastfeeding women, but two safety trials are ongoing with these populations.[2] |
| Instructions for use | Take one pill daily with fluid (e.g. water) | Insert the ring in vagina, wear for one month and then replace |
| | Does not need to be taken with food | |
| Availability | Approved/available in many countries Globally | Pending regulatory approval |
| Manufacturer | Gilead sciences | International Partnership for Microbicides (IPM) |
| Appearance and Size | Single pill 19 mm x 8.5 mm | Flexible silicone ring; 56 mm outer diameter |
| Packaging | A pill bottle (30 pills/bottle) | A sealed single pouch for each ring |
| Colour | Blue | off-white |

[1] https://www.who.int/hiv/pub/toolkits/prep-preventing-hiv-during-pregnancy/en/.

[2] https://www.ipmglobal.org/our-work/our-products/dapivirine-ring.

## Participants

Independent sampling was applied to recruit predominantly PrEP-naïve women who were currently or recently pregnant or breastfeeding from a range of community and clinic settings in the four locations, including street outreach, word of mouth or referrals from community advisory boards, as well as antenatal and postnatal clinics. Inclusion criteria included, being HIV-uninfected (by self-report), aged 18 to 40, currently or recently (in the last two years) pregnant or breastfeeding, proficiency in the local language (Chichewa in Malawi, Zulu or English in South Africa, Luganda in Uganda and Shona in Zimbabwe), and able and willing to provide consent.

## Procedure

All women provided written informed consent prior to participation. Demographic and behavioural information were collected prior to the start of the FGDs through interviewer administrated questionnaires in the relevant local languages. During the FGDs, and prior to discussing the products, women viewed a brief educational video and had the opportunity to handle the prototype of the placebo products (vaginal ring, oral pills; Table 1; [7]).

FGDs ranged from 7–10 participants in size and were conducted in the local language using a guide developed by the research team. Topics discussed included risk perceptions, health-related decision making, key influencers, and interest in HIV prevention methods while P/BF (S1 File). Each FGD was audio recorded, transcribed and verbatim translated into English.

## Analysis

**Primary analysis.** All site research staff joined a two-day workshop for Participatory Rapid Qualitative Analysis and Capabilities Building [33]. In this workshop eight key topics were identified from reading and discussing the transcripts from the women's FGDs. Next, themes and sub-themes were constructed to inform the iterative development of the codebook,

which was further refined by key members of the MAMMA study team. The codebook was applied to guide the thematic analysis of all transcripts, which were analysed in Dedoose software (v7.0.23) with coders meeting weekly to maintain high (80%) intercoder reliability, resolve coding discrepancies and discuss emerging themes [7].

**Secondary analysis—application of the Theoretical Framework of Acceptability.** A hybrid approach of deductive and inductive thematic analysis [34] was applied to complete the secondary analysis of the eight FGDs with P/BF women. First, all 20 grandparent code descriptions within the codebook were reviewed for applicability for the TFA analysis. Seven grandparent codes including parent and child codes (acceptability, barriers ring, facilitators ring, barriers oral pill, facilitators oral pill, prevention methods and risk) were selected to produce code reports, which consisted of the primary data from all eight FGDs.

Next, all data from the seven select code reports was read and reviewed by MS, an expert in applying the TFA. Only five code reports were deemed relevant for the acceptability analysis of both PrEP methods (Acceptability, barriers ring, facilitators ring, barriers oral pill, facilitators oral pill) (S2 File). For each of the five code reports, key steps included (a.) reading participants' utterances within each of the code reports and coding perceptions about both the oral pills and vaginal ring in line with each of the definitions of the seven TFA constructs and (b.) assigning the identified perceptions to one of the TFA constructs.

After all data from the coded reports were analysed into relevant TFA constructs, themes relevant within each of the TFA constructs were inductively generated. In this study, the themes represent the P/BF women's perceptions about the acceptability of both the oral pill and vaginal ring as HIV preventative methods. All themes generated across transcripts within each TFA construct category were discussed with the co-author, AVDS, until agreement was reached. Themes were reworded to convey meaning that represented majority of participant responses, using exact wording (English translations) by the participants whenever possible.

## Results

Across four African sites, 65 P/BF women joined a total of eight FGDs (Table 2). FGD duration ranged from 1 to 3 hours (average 2 hours). Women had experienced a mean of 2.4 pregnancies; 50% of the women reporting they were pregnant at the time of the FGD, and 74% had ever breastfed. All women knew the male condom, but only 45% knew of oral PrEP and 35%

**Table 2. Characteristics of sample by country and overall.**

| Variable | Malawi (N = 15) | South Africa (N = 15) | Uganda (N = 18) | Zimbabwe (N = 17) | Total (N = 65) |
|---|---|---|---|---|---|
| **Mean age (years)** | 26.7 | 28.0 | 27.2 | 26.6 | |
| **Secondary education completed** | 6 (40.0%) | 11 (73.3%) | 4 (22.2%) | 12 (70.6%) | 33 (50.8%) |
| **Married or living with partner** | 14 (93.3%) | 4 (26.7%) | 16 (88.9%) | 16 (94.1%) | 50 (76.9%) |
| **Currently pregnant** | 8 (53.3%) | 6 (40.0%) | 11 (64.7%) | 7 (41.2%) | 32 (50.0%) |
| **Mean pregnancies** | 2.5 | 1.9 | 3.1 | 2.3 | 2.4 |
| **Breastfed** | 12 (80.0%) | 10 (66.7%) | 15 (83.3%) | 11 (64.7%) | 48 (73.8%) |
| **Awareness of HIV preventative measures** | | | | | |
| • Male condoms | 15 (100%) | 15 (100%) | 18 (100%) | 17 (100%) | 65 (100%) |
| • Oral PrEP | 4 (26.7%) | 7 (46.7%) | 10 (55.6%) | 8 (47.1%) | 29 (44.6%) |
| • Vaginal Ring | 5 (33.3%) | 4 (26.7%) | 12 (66.7%) | 2 (11.8%) | 23 (35.4%) |
| **Ever used HIV preventative measures:** | | | | | |
| • Male condoms | 11 (73.3%) | 14 (93.3%) | 16 (88.9%) | 10 (58.8%) | 51 (78.5%) |
| • Oral PrEP | 0 (0.0%) | 3 (20.0%) | 0 (0.0%) | 0 (0.0%) | 3 (4.6%) |
| • Vaginal ring | 0 (0.0%) | 0 (0.0%) | 0 (0.0%) | 0 (0.0%) | 0 (0.0%) |

knew the ring. Most women had used male condoms previously, three reported previous use of oral PrEP and none had previously used the ring.

## Theoretical Framework of Acceptability

Table 3 displays the key themes and sub-themes generated from the TFA analysis reflecting participants prospective acceptability of both HIV PrEP methods. Data was coded into all of

**Table 3. Themes within the TFA construct representing prospective acceptability of oral PrEP and vaginal ring during pregnancy and breastfeeding.**

| Key themes | Affective attitude | Burden | Ethicality | Opportunity costs | Perceived effectiveness | Self-efficacy | Intervention coherence |
|---|---|---|---|---|---|---|---|
| representing each TFA construct | How an individual feel about the intervention | The perceived amount of effort that is required to participate in the intervention | The extent to which the intervention has a good fit with an individual's value system | The extent to which benefits profits, or values must be given up to engage in the intervention | The extent to which the intervention is perceived to be likely to achieve its aim | The participants confidence that they can perform the behaviour(s) required to participate in the intervention | The extent to which participant understands the intervention and how it works |
| Oral pills | (+) Familiarity of taking oral pills | (-) Difficulties to take the daily dose of the pill | (-) Cultural norms discourage pregnant women to take medications | (+/-) Mixed perceptions whether taking the pills will interfere in women's daily lives (e.g. Pills may cause vomiting, dizziness). | (+) The pills will protect mother and baby during pregnancy and breastfeeding from getting HIV | (-) Concerns in ability to take the pills due to size | (-) Taking pills is unsafe for the baby or the pregnancy |
| | (-) Dislike the size of pills | (-) Experiencing potential side effects will be burdensome | | | (+/-) Pills may work for some women and not others | (-) Lack of confidence in remembering to take pills daily | (-) Taking pills during breastfeeding may dry out production of milk |
| | | | | | | | (-) Questions about taking the pills |
| vaginal ring | (+) Discreetness of the ring once inserted | (+) Ease of setting the vaginal ring and forgetting about it for a month | (-) It's taboo to insert things in the vaginal whilst pregnant | (+/-) Using the ring may or may not interfere with women's daily lives (e.g. use may cause mental discomfort) | (+/-) Uncertainties about the effectiveness of the ring | (+) Confidence in using the ring because of its discreetness | (-) Concerns about hygiene when using the ring |
| | (-) Dislike the size and look of the ring | | | | (-) Effectiveness linked to correct placement | (-) Lack of confidence in inserting and removing the ring | (-) Concerns about long term health effects of inserting a ring |
| | | | | | | | (-) fate of ring at time of delivery and harm to the baby |
| | | | | | | | (-) Questions about length of using the ring and protection the ring provides |
| Both PrEP methods | | | (+) Women should have the choice to decide what PrEP methods to use | | | | (+) Clear understanding of routes of protection for both PrEP methods |
| | | | (+) Endorsement from healthcare professionals on taking both PrEP methods is key | | | | (+/-) Mixed understanding about duration of using both PrEP methods |

*(Continued)*

**Table 3.** (Continued)

| Key themes | Affective attitude | Burden | Ethicality | Opportunity costs | Perceived effectiveness | Self- efficacy | Intervention coherence |
|---|---|---|---|---|---|---|---|
| representing each TFA construct | How an individual feel about the intervention | The perceived amount of effort that is required to participate in the intervention | The extent to which the intervention has a good fit with an individual's value system | The extent to which benefits profits, or values must be given up to engage in the intervention | The extent to which the intervention is perceived to be likely to achieve its aim | The participants confidence that they can perform the behaviour(s) required to participate in the intervention | The extent to which participant understands the intervention and how it works |
| Suggested strategy to enhance acceptability | | *General education* | *Involvement of local communities & HCPs* | | *General education* | *Practice and skills building General education* | |
| | | • Testimonials from pregnant or breastfeeding moms | • Involve trusted sources in the intervention roll out, & implementation. | | • Supplement with evidence from PMTCT studies, ongoing trials (when available) | • Provide strategies to enhance self-efficacy e.g. reminders to take pills; teach women how to insert and remove the ring (prior to dispensation) | |
| | | • Describe side effects of PrEP options | • Educate HCPs to ensure support and endorsement | | • Explain mechanism of action | | |
| | | | • Differentiate traditional vaginal practices and ring use | | • Educate about vaginal hygiene and PrEP products | | |

Notes: * **(+)** indicate a positive reflection of the TFA construct (e.g. Affective attitude- *familiarity of taking pills*). **(-)** indicate a negative reflection of the TFA construct (e.g. Burden–*experiencing potential side effects will be burdensome*).

**(+/-)** indicate both a positive and negative reflection of the TFA construct (e.g. Opportunity costs- *Using the ring may or may not interfere with women's other life priorities*).

Acronyms: HCP = health care provider, ARV = antiretroviral medications, PEP = post exposure prophylaxis, PrEP = pre-exposure prophylaxis, TFA = Theoretical framework of acceptability.

the seven TFA constructs for both the pills and the ring. A definition of each of the TFA constructs in the context of this study is summarized in Table 3, and the themes within each construct are discussed below. Each of the women's given names are pseudonyms.

**Affective attitude (P/BF women's positive and negative feelings towards the two PrEP methods).** *Familiarity of taking pills*. Women expressed a positive attitude and likeness towards taking the pills, as they felt a sense of familiarity in the process with this dosage form.

> "*I will be very happy to take this pill to protect me from HIV since I have always taken the vitamin pills every day. In this case I will be protected from HIV so I will be happy*" (Ropa, age 19 Zimbabwe).

*Dislike the size of pills*. A common concern and dislike expressed amongst women was the size of the pills, which they stated was too large, and would be difficult to take daily.

> "*The big size, one can find it difficult to swallow and generally some people do not like large pills.*" (Sarah, 28, Uganda)

*Discreetness of the ring once inserted*. Some women expressed a preference for the ring, as it provides a discreet method for protecting themselves against HIV, and also because it only needed to be changed monthly, in comparison to having to take the pills daily.

"*I would prefer the ring for the following reasons, once inserted one can move about with it, you can have sex with the husband just as you always do without any problem so I think the ring is much better because it is done once in a whole month.*" (Deborah, 34, Malawi)

*Dislike of the size and look of the ring.* Some women however expressed a dislike of the ring, as they felt the material of the ring was too hard and the size was too large to insert into the vagina, and as a result would cause discomfort.

"*I think of discomfort, I just cannot be comfortable because it looks like it's hard so I might feel it inside.*" (Asanda, 26, South Africa).

**Burden (perceived amount of effort [e.g. ease/difficulty; side-effects] required in taking the two PrEP methods).**   *Difficulties to take the pill daily.* Women expressed that taking the pill daily would be difficult in some cases, due to everyday household and work commitments.

"*It can be difficult for you to take your pill daily because sometimes you found that at 8 a.m., you are at work and busy, you have too much work, you cannot go to the kitchen and get yourself some water to take the pill.*"(Ngwanenyana, 26 South Africa).

*Experiencing potential side effects will be burdensome.* A common concern across all FGDs centred on potential side effects associated with the pills. Women anticipated added burden due to side effects such as vomiting and headaches. Not having received information on the potential side effects (the video did not cover side effects) influenced women's decisions on whether they would engage in taking the pills as a preventative method during pregnancy.

"*As a pregnant mother, what first comes into my mind is the issue of side effects to me and the baby, because just like any other pill there are side effects. The other thing is, since my hormones are already tempered around with because of pregnancy, will the PrEP pill go down well with me?*" (Tanya, 31, Zimbabwe)

*Ease of setting the ring and forgetting about it for a month.* Many participants felt using the ring would be easier, as there would be no need to remember to take it daily. The ring would only need to be changed monthly, which would suit some women's lifestyles better.

"*The pill can be easily forgotten and the big size, one can find it difficult to swallow. With the ring once inserted you will not have any problems. Once inserted one will wait till end of the month.*" (Sarah, 34, Zimbabwe)

**Ethicality (the extent in which the two PrEP methods are perceived as having a good fit with an individual's value system, and local norms).**   *Cultural norms discourage pregnant women to take medications.* Participants expressed that their cultural norms discouraged women from taking any form of medication whilst pregnant to avoid harm to the baby.

"*. . . you don't know how the drugs will work in your body while you are pregnant. They may come with so much strength that may lead to fatigue or can even destroy the baby you are expecting. So, yes, it is good that the drugs will protect from HIV, but they may bring some undesirable side effects while you are pregnant; as it is said that when one is pregnant, she should not be taking drugs.*" (Lucy, 29, Malawi)

*It's taboo to insert things in the vagina whilst pregnant.* A minority of women across all FGDs stated that their cultural norms deemed it taboo to insert anything into their vagina during pregnancy. For those, it was not acceptable to use the ring during pregnancy:

"*I think it may be taboo because people will not understand you inserting things in the vagina while pregnant. You can only drink things, so when you insert things in the vagina is just something else.*" (Vanessa, 23, Uganda)

Women also said that inserting anything into the vagina during pregnancy could cause harm to the foetus.

"*I think it's not right while pregnant because you will have to always insert your fingers every month, maybe you will never know whether you are hurting the foetus (Pink, 22, South Africa)*

*Women should have the choice to decide what PrEP methods to use.* Women advocated for having the choice in deciding which PrEP method they may decide to use, and when they would start using one of the methods during pregnancy and/or lactation.

"*I think women will choose what they want to use, I think you are not going to choose for them, they will have to choose, and at what time they want to use these products.*" (Makhosi, 40, South Africa)

*Endorsement from healthcare professionals on taking both PrEP methods is key.* Women expressed that their decision to take either of the PrEP methods would be dependent on the advice provided by their healthcare professionals and would only take either of the methods if prescribed by their medical provider.

"*I have said before that I don't take any medication that is not prescribed by my doctor. So, I don't think I would be interested in using it unless it is prescribed by my doctor that I must use.*" (Ngwanenyana, 26, South Africa)

**Opportunity costs (whether taking using the PrEP methods would interfere with other important priorities in the daily lives of P/BF women).** *Mixed perceptions whether taking the pill will interfere in women's daily lives.* Some women felt taking the pills was a priority in protecting themselves and their babies against HIV, thus it was up to each individual to ensure they remember to take the pill daily, especially during pregnancy.

"*I don't think it will interfere in a bad way because after all it's not a stress taking a pill, you just take a pill and swallow it that's it.*" (Lisa, 27, South Africa).

Some women, however, expressed concerns about the implications of the pills' side effects on their daily lives, such as vomiting and dizziness, which would be disruptive and exacerbate pregnancy related symptoms.

"*The family planning pill, I have been taking them for 8 years. Every morning I would wake up vomiting like someone who is pregnant. So, when you take this pill for a long time, won't it cause the same things in your life? Such things can now affect your daily life because in the morning you can fail to do your daily chores feeling dizzy or something*" (Jane, 30, Zimbabwe)

*Using the ring may or may not interfere with women's' daily lives.* Similarly, there were mixed opinions about the ring. Some women felt the ring would not interfere in their daily lives, as it only needed to be changed monthly.

"*If the vaginal ring is inside and people are able to forget about it, it cannot interfere. Even the tampon is worse because it gets full and you can feel it is full but the vaginal ring stays there for a month.*" *(Lisa, 27, South Africa)*

Whereas, other felt that the ring would cause disturbances in their daily lives, as it was a new method and women may not feel comfortable knowing the ring has been inserted, and as an unfamiliar vaginal product it may generate mental discomfort.

"*I feel the ring can disturb your daily life, because it is a new thing in your life. When you have inserted that thing, especially if it's at the beginning, from time to time, you may feel like it has come out or keep asking yourself if it's in place or if it has moved every time you have had sex. You can somehow be in a disturbed state because you are not yet used to it.*" *(Lucy, 29, Malawi).*

**Perceived effectiveness (perceived protection against HIV by the two PrEP methods).**
*The pills will protect mother and baby during pregnancy and breastfeeding from getting HIV.*
Women felt that oral PrEP would be an effective method in protecting themselves against contracting HIV.

"*I will know that I will stay safe [HIV negative] even when my partner engages in sex with other women because it will not be possible for him to infect me. . .. I would take it because it would stop me from worrying about getting infected with HIV.*" *(Sarah, 34, Zimbabwe).*

Women also felt that oral PrEP would confer protection to them and the baby during pregnancy, and breastfeeding.

"*I feel this method of taking oral tablets is very helpful because when you are pregnant, you will not be worried of getting infected at any time. You will know that you are protected, together with the child you are expecting.*" *(Favour, 29, Malawi)*

*The pills may work for some women and not for others.* Some women expressed concerns with regards to the perceived effectiveness of oral PrEP, in particular that the pills would be effective in protecting against HIV for some women and not for others.

"*What if it treats another person okay and with me it doesn't, maybe she can take it and when I take it I have side- effects even though they know that its best it doesn't mean it's going to work for everyone.*" *(Grey, 26, South Africa).*

Women expressed that effectiveness of oral PrEP in pregnancy would be unique to each woman. Their main concern of oral PrEP not being effective was due to past sickness and frequent vomiting in their own pregnancies.

"*It might not work for me because during pregnancy I vomit a lot and that makes me doubt whether it can stay inside when I vomit.*" *(Vanessa, 23, Uganda)*

*Uncertainties of the effectiveness of the ring.* Women discussed uncertainties with regards to the effectiveness of the ring given its local (as opposed to systemic) drug exposure.

"*The ring protects on the outside so that it doesn't spread on the whole body, what if you get like a blood infection?" (Mpho, 27, South Africa)*

Other uncertainties related to women's cultural concerns about inserting anything into the vagina during P/BF, as the ring may be unhygienic and cause infections or other health conditions.

"*I wonder whether it doesn't have side effects. . .the other is that it [the ring] is so hard and it is supposed to be inserted inside the vagina and how you can keep it clean when you remove it out and then reinsert it?" (Barbara, 36, Uganda).*

*Effectiveness linked to correct placement.* Other women also felt that the positioning of the ring may have consequences for its effectiveness in protecting against HIV.

"*We want to know whether this thing [Ring] will not move from its position while inside the vagina. Shifting might affect ring effectiveness." (Nyasha, 22, Zimbabwe).*

**Self-efficacy (perceived confidence in using each of the PrEP method).** *Concerns in ability to take pills due to size.* Women expressed concerns in having the confidence to take the pills due to their size. Many women felt that pills themselves were too large, thus they may feel the pill as they swallow it, or the pill may get stuck in their throats.

"*I am saying the size of the pill is too big. I think it's too big to swallow and you might feel it, no ways." (Pink, 22, South Africa).*

*Lack of confidence in remembering to take pills daily.* Women felt that they may forget to take the pill daily, thus would not work in protecting themselves from HIV.

"*I think you might forget because sometimes you would sleep out, and it would be hard to leave someone's house you are visiting. . .and then maybe you didn't take them with you, or you forgot them at home." (Grey, 26, South Africa)*

*Confidence in using the ring because of its discreetness.* Women acknowledged the advantages of discretion in using the ring, with some women indicating that they felt more confident in using the ring in comparison to the pill, as they felt their partners would disapprove or interfere with their decision to protect themselves against HIV.

"*People will say, "Ah, what are these pills for?" and they can actually say that you have HIV. The ring is discreet. If husbands are not cooperative, women can just use it as long as it will not be felt during sex" (Tanatswa, 25, Zimbabwe)*

*Lack of confidence in inserting and removing the ring.* Fears about the ring were expressed, with many women being worried about the size of the ring, and how they would insert and remove the ring themselves and place it correctly.

"*I fear inserting it myself because I might insert it wrongly and it goes to a different part from where it is supposed to go. I feel I might not be able to insert it the way a health worker would have done it (Suzan, 29, Uganda)*

**Intervention coherence (understanding of how taking each PrEP method works in protecting against HIV, based on information provided).** Many of the themes identified below, indicate women in this sample had a poor understanding of how the two PrEP methods would work in protecting against HIV, and expressed concerns that using them could result in negative consequences for the baby. This highlighted the limitations of the short educational video and product handling opportunity provided during the FGDs.

*Taking pills feels unsafe for the baby or the pregnancy.* In several FGDs women expressed concerns about the safety of using the pills during pregnancy, specifically that it may lead to the baby being born with disabilities or that it could lead to a miscarriage.

"*The first thing I think about is whether it won't affect the baby because you have told us that we can take it during pregnancy, but won't it affect the baby, will the baby be disabled?*"
*(Agatha, 21, Uganda)*

*Taking pills during breastfeeding may dry out production of milk.* Some women believed taking oral PrEP whilst breastfeeding would provide protection, and one specifically articulated clearly the risk of maternal transmission through breastmilk.

"*Since the baby is breastfeeding, he may get infected through the mother's breast milk. So, this is worrisome to the mother. So, it would be very helpful to use the oral tablets method while breast feeding, in order to protect yourself as well as the baby from getting infected. (Kheliwe, 25, Malawi).*

However, common considerations also focused on how pills can affect their milk production, resulting in side effects for their baby.

"*During breastfeeding, you would not know if they will not affect the milk or cause side effects on the breastfeeding baby. So, I think it's a bit tricky using the PrEP pill when breastfeeding.*"
*(Charlene, 19, Zimbabwe)*

*Questions about taking the pills.* Women also had several questions about the use of both PrEP methods. Questions about the pills centred around taking the pills with food, what would happen if they were to miss a dose and if the PrEP pills worked in similar way to the contraceptive pills.

"*Do they [taking the oral pills] require someone to eat well like how HIV patients are told to do? In case I missed a day, does that have an effect on me and do I have to take it on a specific time like how family planning pills are required?*" *(Suzan, 29, Uganda)*

*Concerns about hygiene when using the ring.* Women across all focus groups expressed concerns about the ring remaining inserted for a month. Women perceived that the use of the ring was unhygienic, as social norms within the community consider the vagina a "dirty" environment.

"*Won't dirt accumulate where it will be placed or when the dirt accumulates, are we going to have our uterus cleaned? Where exactly will the ring be placed? Will the ring be clean where it will be placed? When releasing the drug for the prevention of HIV, will it also be releasing the dirty that would have accumulated on it or the dirty will just be kept there?*" (Tendai, 37, Zimbabwe).

Vaginal insertion of a device during pregnancy was considered against social norms, and possibly against medical advice as this could cause harm to the baby.

"*Some doctors do say if you insert some things here, those things do affect the head of the baby and the baby might become slow and the baby's eyes might have a discharge, and it can lead do wounds and a baby will end up blind.*" (Red, 30, South Africa).

*Concerns about long term health effects of inserting a ring.* Women expressed concerns about long term use of the ring, and potential health effects associated with inserting the ring.

*Where I feel the ring can give us problems, is because some people say it is not good to be inserting fingers/things on the cervix as this may cause cancer or introduce bacteria.*" (Lucy, 29, Malawi)

*Fate of ring at time of delivery and harm to the baby.* Women did not understand what would happen to the ring at the time of delivery, specifically whether the ring would be removed before or during their delivery.

"*Before you deliver, they do check you- how many centimetres, obviously, when they put a finger in, they will feel the ring? This thing is a rubber, the baby's head is here, and the baby is coming and pushing it and the ring does it slide on the side as baby pushes out?*" (Juicy lips, 36, South Africa).

*Questions about length of using the ring and protection the ring provides.* Questions about the ring also centred around conditions for and duration of protection.

"*If I have used it for say three years and then one time I do not have it and I happen to have sex with a man when it is not inserted, do I get infected with HIV or is the medicine is still in my body?* (Ropa, 19, Zimbabwe)

*Clear understanding of routes of protection for both PrEP methods.* Many women understood the routes of protection of both PrEP methods based on the information they had viewed in the short video. Women explained the pills provided protection against infection during multiple routes of exposure.

"*These pills can help for both oral and vaginal sex because there are those who prefer oral sex to vaginal sex.*" (Grey, 26, South Africa)

Women understood that the pills have been designed to prevent HIV by being in control of their own health, especially whilst pregnant.

"*These tablets can really help us because men do whatever they want when we are pregnant. They take advantage to go out with other women. So, if you use these tablets to protect yourself*

*from HIV, then you and your unborn child will be protected from acquiring the virus." (Lucy, 22, South Africa).*

Women also understood that the ring was to be changed monthly, and the key was to remember when to change it.

*"This vaginal ring I feel is a good method because when you put it once, it means it is there for the whole month and you will be using that very same ring, while with tablets you may forget due to being occupied." (Favour, 29 Malawi)*

*Mixed understanding about duration of using both PrEP methods.* Some women did not understand the purpose of the pills, or duration of use and had concerns about starting and stopping the use of the pills, for example believing that their immune systems would begin to weaken, and they no longer would be protected against HIV.

*"I heard that when the blood gets used to medicine it weakens [immunity reduces] if one stops taking that medicine. So, that would mean that in case I start taking that medicine and then stop my immunity will also reduce where it can't fight off the virus. That would mean that I have to take it forever just like HIV positive patients?"(Samantha, 30, Uganda)*

Some women also did not understand how the ring would be inserted and removed.

*"I just don't understand it. How do you remove it, what if you can't, what if you inserted it wrongly and it affects you?" (Nonhlanhla, 34, South Africa).*

In summary application of the TFA constructs to this set of qualitative data highlighted this group's interest in new biomedical HIV prevention approaches but also a range of areas that could be strengthened to increase prospective acceptability. Based on the negative and some of the neutral/ambivalent themes generated within five of the TFA constructs (burden, ethicality, perceived effectiveness, self-efficacy and intervention coherence) we generated insights that intervention developers can focus their efforts on to enhance the acceptability of each PrEP method in the future (bottom row, Table 3).

## Discussion

To our knowledge this is the first qualitative study to apply the TFA to explore prospective acceptability of two novel HIV PrEP products, the daily oral pills and the monthly vaginal ring among P/BF women in sub-Saharan Africa. The findings from this study suggest, applying the TFA as a framework of analysis provided useful insights about the prospective acceptability of both PrEP methods and how acceptability of each may be enhanced through specific strategies.

We identified mixed perceptions amongst study participants with regards to the prospective acceptability of both PrEP approaches. Women highlighted the familiarity of taking pills, understanding the purpose of taking pills, and the perception that oral PrEP is an effective method to protect themselves and their baby from HIV during pregnancy and breastfeeding. With regards to the ring, women emphasized the ease of using the ring given low burden on the user and liking the discrete and effective method of protection against HIV due to its monthly duration and invisibility once in place. For both PrEP methods, women expressed choice of method and personal preference was key to selecting which product to personally use.

In line with previous research focusing on microbicide acceptability, our findings indicate the physical functions of both PrEP products (e.g. size of oral pills, size of ring, feeling of ring) can provide some insights into intention to use either PrEP product in the future [26, 28]. Furthermore, our findings highlight the importance of considering environmental, social and cultural norms, and their influence on women's perceptions of acceptability. Specifically, women's cultural norms discourage pregnant women to take medications, and women viewed inserting the ring in the vagina whilst pregnant as taboo.

The TFA provided a comprehensive framework to guide our analysis of women's prospective acceptability of the oral pills and vaginal ring. Data from all FGDs was coded into all seven of the TFA component constructs. However, it was challenging to code data that focused on women's thoughts on the potential side effects, and safety concerns associated with the pills. Women's perspectives on side effects centred around the potential for oral PrEP to cause side effects similar to the contraceptive pill. Concerns relating to the safety of oral PrEP were reflected as women's misunderstanding of how that pill works. Women stated that taking the oral pill could lead to miscarriages or that the baby may be born with disabilities. This issue highlights the importance of understanding the nuances of the TFA definitions. Through iterative discussions, data referring to the potential side effects anticipated with the pills was coded into the construct of burden, on the basis, that each intervention is likely to have potential benefits and potential disadvantages. Here, perceptions that women may experience side effects (e.g. vomiting) were considered a high burden (i.e. high levels of effort) and a disadvantage of taking oral PrEP. Data referring to anticipated safety concerns with taking the pills was coded into the construct of intervention coherence (i.e. poor understanding of how oral PrEP works).

Our analysis revealed aspects of both PrEP approaches that could be modified to enhance perceptions of acceptability. The construct of intervention coherence included a number of themes, centring around women's understanding of how both PrEP methods work, and their uncertainty of the risks associated with each method. In the MAMMA study, the information provided in the educational video, was (by design) limited. A key strategy to address the majority of negative themes, is to provide more thorough general educational about each PrEP approach, including mechanism of action and side effects. For oral PrEP, information should include the safety of taking the pills during pregnancy and breastfeeding (i.e. taking the pills will not dry out milk production or cause harm to the baby). For the ring, information should focus on vaginal hygiene, ring placement and concerns about the whereabouts of the ring at delivery.

For patients in future PrEP implementation settings, offering product choice with suitable educational tools to decide among available options will be important. Such tools have supported women in making decisions about methods of contraception, and are also being developed for PrEP [35, 36]. On a practical level, in-person demonstration of insertion and removal of the ring using pelvic models, diagrammatic representations or visual aids and/or in-clinic practice can build user confidence and skills [37].

Demonstration of pill swallowing techniques can facilitate those with pill taking problems [38]. Women also suggested seeing testimonials from product ambassadors i.e. P/BF women that have used these methods [7], to help new users view both methods more favourably, specifically by hearing about 'real' women's experiences and their reasons for choosing PrEP.

The main limitation of this study was the small number of FGDs per participant per site, as we didn't intend to do cross country comparisons at the outset. The study was intended for overall analysis in high HIV prevalence and incidence settings, and we found many similarities in responses across the study sites. Some site variations were previously noted, for example with respect to P/BF women's differing views about the effects of taking bitter medications, or

regarding the ring being possibly misperceived by providers as an abortion tool [7]. Because the study was not designed to assess attitudes by geographical or cultural settings, more research would be needed to ascertain whether these relate or not to cultural norms.

Second, whilst the TFA was applied to complete the secondary analysis of the data, the FGD guide was not informed by the TFA. Thus, none of the questions or topics discussed purposively explored any of the TFA constructs. Nevertheless, the structure of the FGDs and open probing ensured participants were able to express their thoughts, experiences and perceptions. As a result, our analysis included a range of themes for all seven TFA constructs about women's' prospective acceptability of both PrEP methods and provided guidance on where to focus efforts to enhance future acceptability.

## Conclusion

This secondary analysis indicates that TFA can be applied to provide a comprehensive assessment of prospective acceptability. Data was analyzed into all seven TFA constructs and provided an indication on how acceptability of both PrEP methods could be enhanced by focusing on perceptions of the end users (i.e. the women) and not just the products themselves. This approach has added value and provided insights into how to refine future intervention materials and plans for implementation.

## Supporting information

**S1 File. MTN-041 pregnant and breastfeeding women Focus Group Discussion (FGD) topic guide.**
(DOCX)

**S2 File. MTN-041 pregnant and breastfeeding women Focus Group Discussion (FGD) topic guide code reports.**
(DOC)

## Acknowledgments

**We would like to acknowledge the entire MTN-041 study team:** Leadership: Ariane van der Straten, Women's Global Health Imperative (WGHI) RTI International (Protocol Chair); Petina Musara, University of Zimbabwe College of Health Sciences Clinical Trials Research Centre (Protocol Co-chair); Julia Ryan, Women's Global Health Imperative (WGHI) RTI International (Research Public Health Analyst); Jeanna Piper, US National Institute of Allergy and Infectious Diseases (Medical Officer); Teri Senn, National Institute of Mental Health (NIMH Chief); Nicole Macagna, FHI 360 (Clinical Research Manager); Study sites, site Investigators of Record and site teams: Malawi, Blantyre site (Johns Hopkins Project-College of Medicine, University of Malawi): Frank Taulo South Africa, Johannesburg site (Wits Reproductive Health and HIV Institute): Krishnaveni Reddy Uganda, Kampala site (Makerere University–Johns Hopkins University Research Collaboration): Juliane Etima Zimbabwe, Chitungwiza, Zengeza site (University of Zimbabwe College of Health Sciences Clinical Trials Research Centre): Petina Musara. Data management, coding and analysis was provided by the Women's Global Health Imperative Program (RTI International, Berkeley, CA): Julia Ryan, Mary-Kate Shapley-Quinn and Zoe Duby (Desmond Tutu HIV foundation).

## Author Contributions

**Conceptualization:** Mandeep Sekhon, Ariane van der Straten.

**Formal analysis:** Mandeep Sekhon.

**Funding acquisition:** Ariane van der Straten.

**Investigation:** Ariane van der Straten.

**Methodology:** Mandeep Sekhon, Ariane van der Straten.

**Resources:** Ariane van der Straten.

**Supervision:** Ariane van der Straten.

**Validation:** Ariane van der Straten.

**Writing – original draft:** Mandeep Sekhon, Ariane van der Straten.

**Writing – review & editing:** Mandeep Sekhon, Ariane van der Straten.

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
