## [Decision Letter · Decision Letter 0]

24 Jun 2021

PONE-D-21-07808

Pregnant and breastfeeding women's prospective acceptability of two biomedical HIV prevention approaches in Sub Saharan Africa: a multisite qualitative analysis using the Theoretical Framework of Acceptability

PLOS ONE

Dear Dr. Sekhon,

Thank you for submitting your manuscript to PLOS ONE. After careful consideration, we feel that it has merit but does not fully meet PLOS ONE’s publication criteria as it currently stands. Therefore, we invite you to submit a revised version of the manuscript that addresses the points raised during the review process.

We look forward to receiving your revised manuscript.

Kind regards,

Karine Dubé, DrPH

Academic Editor

PLOS ONE

Journal Requirements:

"The authors have declared that no completing interests exist. "

We note that one or more of the authors are employed by a commercial company: ASTRA Consulting.

2.1. Please provide an amended Funding Statement declaring this commercial affiliation, as well as a statement regarding the Role of Funders in your study. If the funding organization did not play a role in the study design, data collection and analysis, decision to publish, or preparation of the manuscript and only provided financial support in the form of authors' salaries and/or research materials, please review your statements relating to the author contributions, and ensure you have specifically and accurately indicated the role(s) that these authors had in your study. You can update author roles in the Author Contributions section of the online submission form.

2.2. Please also provide an updated Competing Interests Statement declaring this commercial affiliation along with any other relevant declarations relating to employment, consultancy, patents, products in development, or marketed products, etc.  

3. We noted in your submission details that a portion of your manuscript may have been presented or published elsewhere.

"The primary analysis of the data set (including a wider range of focus groups and participants – men, and family members of pregnant and breastfeeding women) has been published elsewhere:

•van der Straten, A., Ryan, J. H., Reddy, K., Etima, J., Taulo, F., Mutero, P., ... & MTN‐041/MAMMA Study Team. (2020). Influences on willingness to use vaginal or oral HIV PrEP during pregnancy and breastfeeding in Africa: the multisite MAMMA study. Journal of the International AIDS Society, 23(6), e25536.

The focus of the primary data set was to explore participants influences and their willingness to use vaginal or oral HIV prep.  In the current study we completed a secondary qualitative analysis, for focus group discussions only with pregnant and breastfeeding women, exploring P/BF women’s prospective acceptability of the daily oral pills and the monthly vaginal ring, by applying the Theoretical Framework of Acceptability to guide the secondary analysis."

Please clarify whether this publication was peer-reviewed and formally published. If this work was previously peer-reviewed and published, in the cover letter please provide the reason that this work does not constitute dual publication and should be included in the current manuscript.

Reviewers' comments:

Reviewer's Responses to Questions

**Comments to the Author**

1. Is the manuscript technically sound, and do the data support the conclusions?

Reviewer #1: Yes

Reviewer #2: Yes

2. Has the statistical analysis been performed appropriately and rigorously? 

Reviewer #1: Yes

Reviewer #2: N/A

3. Have the authors made all data underlying the findings in their manuscript fully available?

Reviewer #1: Yes

Reviewer #2: No

4. Is the manuscript presented in an intelligible fashion and written in standard English?

Reviewer #1: Yes

Reviewer #2: Yes

5. Review Comments to the Author

Reviewer #1: This study was a secondary analysis embedded within the Microbicide Trial Network (MTN)- funded Microbicide/PrEP Acceptability among Mothers and Male Partners in Africa (MTN-041/MAMMA). The parent study was a multi-site qualitative study across 4 African countries to identify individual, interpersonal, social and cultural factors that may affect the uptake by pregnant and breastfeeding women in Africa of the monthly dapivirine vaginal ring (a product that has received prequalification from WHO, and is under regulatory consideration in multiple settings), and a pill for daily oral pre-exposure prophylaxis (PrEP) approved for use by women globally and endorsed by WHO. Acceptability was assessed using a recent Theoretical Framework of Acceptability (TFA) that has been developed to facilitate both quantitative and qualitative assessments of healthcare intervention acceptability. The TFA was developed by synthesizing the findings from reviews and applying deductive methods to theorize the concept of acceptability. The TFA consists of seven component constructs: Affective attitude, Burden, Perceived effectiveness, Ethicality, Intervention coherence, Opportunity costs and Self-efficacy. Of note, the focus group discussion guide used in the study had not been informed by the TFA, which was applied to the codes and output post hoc; this is noted in the study’s limitations.

Overall, the study was thoughtfully conceptualized and it addresses an area in which there is very little research. HIV prevention during pregnancy and lactation is very important, as these are times of high HIV susceptibility; but PrEP is underutilized during these times in a woman’s life. Additionally, little is known about awareness, knowledge and attitudes about PrEP during pregnancy and lactation, and social and behavioral determinants of PrEP use during these periods are poorly understood. As such, the study is an important step towards gaining a better understanding, with the goal of increasing uptake of this highly efficacious intervention during pregnancy and breastfeeding. An additional strength of the study is that it explores acceptability as a multifactorial concept, including not only product attributes, but also user, partner and community factors. Applying the TFA construct in this realm is a methodological advance. Inclusion from women from four different countries increases generalizability of findings, even though the number of women from each site was relatively small. The study found that there was variable awareness at the different sites of the two PrEP options during pregnancy/lactation (as low as 26% for oral PrEP awareness in one of the sites and only 11% awareness of the ring at one site). The findings also highlighted the importance of personal beliefs and societal/cultural norms as determinants of product acceptability, particularly during pregnancy and breastfeeding, and suggested factors that are important to include in informational and counseling materials of these PrEP modalities when they are being implemented.

Specific comments:

The document does not have page and line numbers, thus it is difficult to specify exact location for each specific comment below:

In Introduction: The statement “Primary findings indicate that there was consensus amongst men and women, that P/BF women are at higher risk of HIV due to their partners infertility.” What does that mean? Why are the partners of pregnant women infertile and why is this relevant to the women’s higher risk for HIV? Please clarify.

Reviewer #2: Pregnant and breastfeeding women's prospective acceptability of two biomedical HIV prevention approaches in Sub Saharan Africa: a multisite qualitative analysis using the Theoretical Framework of Acceptability

Summary of review:

Thank you for the opportunity to review this manuscript which describes a qualitative exploration of the prospective acceptability of oral PrEP and a vaginal ring PrEP. The authors have clearly and rigorously performed a secondary analysis to tie PrEP usage themes to the TFA. This is a thorough analysis on a concept that has received little attention previously. I have only a few minor comments including potentially incorporating differences among the participants at different sites since the four countries will likely have differing norms and values in regards to pregnancy, breastfeeding, and HIV prevention.

Additional questions and suggestions are offered by manuscript section:

Introduction

1. Paragraph 2, Line 4, please spell out acronyms prior to use (TC/TDF).

2. Paragraph 6, Line 2, use of study twice in the same sentence, “the Microbicide Trial Network (MTN)- funded a study multi-site qualitative study.”

3. Paragraph 6, Line 2, please explain the primary findings from reference 7, indicating that participants felt P/BF women were at higher risk of HIV due to partners’ infertility. Was infertility an inclusion criteria for study participation?

4. Final paragraph, Line 2, in the sentence, “In this paper, we apply the TFA to complete a secondary analysis of the eight FGDs completed with B/BF as part of the MAMMA study” B/BF is likely a typo and should be P/BF.

Discussion

1. Paragraph 4, line 3-4, can the authors expand upon why it was difficult to code data on side effects and safety concerns?

2. Given that the interviews were conducted in four very different countries and settings, could the authors include some discussion of the variability in cultural norms and practices and their effect on the acceptability of PrEP and the vaginal ring?

6. PLOS authors have the option to publish the peer review history of their article (what does this mean?). If published, this will include your full peer review and any attached files.

Reviewer #1: No

Reviewer #2: No

---

## [Author Response · Author response to Decision Letter 0]

23 Jul 2021

Please see Response to Reviewers letter attached as part or revised submission.

---

## [Editor Report · Decision Letter 1]

27 Oct 2021

Pregnant and breastfeeding women's prospective acceptability of two biomedical HIV prevention approaches in Sub Saharan Africa: a multisite qualitative analysis using the Theoretical Framework of Acceptability

PONE-D-21-07808R1

Dear Dr. Sekhon,

We’re pleased to inform you that your manuscript has been judged scientifically suitable for publication and will be formally accepted for publication once it meets all outstanding technical requirements.

Kind regards,

Karine Dubé, DrPH

Academic Editor

PLOS ONE

Additional Editor Comments (optional):

N/A

Reviewers' comments:

N/A

---

## [Editor Report · Acceptance letter]

5 Nov 2021

PONE-D-21-07808R1 

Pregnant and breastfeeding women’s prospective acceptability of two biomedical HIV prevention approaches in Sub Saharan Africa: a multisite qualitative analysis using the Theoretical Framework of Acceptability 

Dear Dr. Sekhon:

I'm pleased to inform you that your manuscript has been deemed suitable for publication in PLOS ONE. Congratulations! Your manuscript is now with our production department. 

Kind regards, 

on behalf of

Dr. Karine Dubé 

Academic Editor

PLOS ONE